# Interface Material Modification to Enhance the Performance of a Thin-Film Piezoelectric-on-Silicon (TPoS) MEMS Resonator by Localized Annealing Through Joule Heating

**DOI:** 10.3390/mi16080885

**Published:** 2025-07-29

**Authors:** Adnan Zaman, Ugur Guneroglu, Abdulrahman Alsolami, Liguan Li, Jing Wang

**Affiliations:** 1Microelectronics and Semiconductor Institute, King Abdulaziz City for Science and Technology (KACST), Riyadh 11442, Saudi Arabia; 2Department of Electrical Engineering, University of South Florida, 4202 E. Fowler Avenue, Tampa, FL 33620, USAliguan@usf.edu (L.L.)

**Keywords:** thin-film piezoelectric-on-silicon (TPoS) resonator, piezoelectric, anchor-related losses, *Q*-factor, Joule heating, localized annealing

## Abstract

This paper presents a novel approach employing localized annealing through Joule heating to enhance the performance of Thin-Film Piezoelectric-on-Silicon (TPoS) MEMS resonators that are crucial for applications in sensing, energy harvesting, frequency filtering, and timing control. Despite recent advancements, piezoelectric MEMS resonators still suffer from anchor-related energy losses and limited quality factors (*Qs*), posing significant challenges for high-performance applications. This study investigates interface modification to boost the quality factor (*Q*) and reduce the motional resistance, thus improving the electromechanical coupling coefficient and reducing insertion loss. To balance the trade-off between device miniaturization and performance, this work uniquely applies DC current-induced localized annealing to TPoS MEMS resonators, facilitating metal diffusion at the interface. This process results in the formation of platinum silicide, modifying the resonator’s stiffness and density, consequently enhancing the acoustic velocity and mitigating the side-supporting anchor-related energy dissipations. Experimental results demonstrate a *Q*-factor enhancement of over 300% (from 916 to 3632) and a reduction in insertion loss by more than 14 dB, underscoring the efficacy of this method for reducing anchor-related dissipations due to the highest annealing temperature at the anchors. The findings not only confirm the feasibility of Joule heating for interface modifications in MEMS resonators but also set a foundation for advancements of this post-fabrication thermal treatment technology.

## 1. Introduction

Microelectromechanical systems (MEMS) piezoelectric resonators have been attracting ever-growing employment across several applications such as sensing [1,2], energy harvesting [3], and frequency filtering and timing control applications [4,5,6]. One of the critical performance metrics of the MEMS resonators, regardless of their transduction mechanism, is the quality factor (*Q*), which corresponds to the ratio of the energy stored to energy dissipated per cycle [7]. Researchers have been investigating different methods to improve the resonator *Q*, such as mechanical coupling isolation by using a phononic bandgap structure [8,9]. Another key parameter is the motional resistance that determines the ability of the resonator to match with the typical 50 Ω characteristic impedance of other RF components. Several attempts have been reported to reduce the motional resistance both in piezoelectrically and capacitively transduced MEMS resonators.

One approach in prior works was to design and implement periodic phononic crystals, through which the acoustic MEMS resonator is anchored to the surrounding substrate [10]. For capacitively transduced resonators, connecting the vibrating resonator body at its corners and nodal locations to reduce motional resistance [11] using solid dielectric material instead of an air gap between the vibrating body and the capacitive electrode [12] has been explored. Alternatively, the Joule heating method has been used to locally anneal the chosen MEMS resonators to improve the overall performances of capacitive, piezoresistive, and piezoelectric devices [13,14,15]. One of the very first attempts conducted via local annealing of capacitive resonators with DC currents to redistribute/remove defects in nickel comb-drive structure has been reported in [13]. Another study was carried out in a double-ended tuning fork (DETF)-based piezoresistive resonator, where the DETF was fabricated out of single-crystal silicon, and the induced heat changed thermophysical properties of the silicon such as the Young’s modulus and heat capacity, which in turn result in improved *Q* [14]. Another investigation was also performed for piezoelectrically transduced nickel resonators with the resonator body made of plated nickel instead of silicon. Researchers locally annealed the nickel layer to enhance the properties and remove the defects from the material [15].

Despite ongoing advancements in piezoelectric MEMS resonator technologies, key limitations such as anchor-related energy dissipation, limited *Q*-factor at high frequencies, and motional resistance mismatch remain significant barriers to their broader adoption in high-performance applications. Previous efforts have focused on solutions like phononic crystal anchoring, mechanical mode coupling, and electrode reconfiguration, each offering varying degrees of success but often introducing complexity in the design or fabrication [16,17,18,19,20,21]. For instance, mode coupling techniques improve the *Q*-factor but require precise structural tuning, while phononic crystals demand intricate patterning and may not suit all device geometries. Electrostatic and mechanical tuning mechanisms, though effective in frequency adjustment, typically do not address anchor-related losses or long-term stability. Moreover, most of these techniques involve sophisticated fabrication processes, added design constraints, or scalability challenges, limiting their practical deployment in wafer-scale manufacturing. Recent studies [22,23,24] have explored localized thermal treatments, but a systematic approach to correlate interface modifications by annealing with enhanced acoustic properties in piezoelectric-on-silicon MEMS remains limited. In contrast, our approach introduces a post-fabrication interface modification using localized Joule heating that targets material properties at the anchor region, enhancing the *Q*-factor and reducing motional resistance without altering the device geometry or fabrication process. This method not only simplifies the integration with existing MEMS device technologies but also enables tunable performance improvements via controlled thermal treatment.

In this study, metal diffusion and platinum silicide formation have been identified as a viable post-fabrication treatment mechanism to improve the quality factor of the thin-film piezo-on-silicon (TPoS) resonators through the Joule heating effect. When a DC current is applied to the resonator through its anchors for a chosen duration, and the temperature of the resonator is elevated to a level that facilitates metal silicide formation between the bottom electrode and the silicon resonator body, at this elevated temperature, the metallic bottom electrode will diffuse into the silicon resonator’s body, altering both the stiffness and density. Consequently, the effective acoustic velocity of the resonator is altered, thus causing a slight frequency shift. Higher stiffness due to the localized annealing enables the resonator to store more acoustic energy, which in turn also enhances the resonator’s *Q*-factor. Meanwhile, the side-supporting anchors reached the highest temperature during all localized annealing conditions, which lowers acoustic energy dissipations through anchor-related losses. To the best of our knowledge, this is one of the first reports that correlates Pt-Si surface interface modification with the piezoelectric MEMS resonator’s performance enhancements in *Q*, motional resistance, and insertion loss.

## 2. Fabrication and Operation

### 2.1. Piezoelectric ZnO-on-Si Resonator Microfabrication Process Flow

Low motional resistance and a high quality factor are two of the main goals when designing and fabricating piezoelectrically transduced resonators. Figure 1a–h present the step-by-step fabrication process flow by showing the perspective, cross-sectional, and top-view diagrams of the piezoelectric ZnO-on-Si resonator during its microfabrication. A strategic design of the top electrode pattern is needed to match the strain field induced in the ZnO piezoelectric transducer layer. Piezoelectric MEMS resonators were fabricated using a silicon-on-insulator (SOI) wafer with a 10 μm thick silicon device layer and a 2 μm thick buried oxide (BOX) layer, as shown in Figure 1a. Initially, the wafer goes through a solvent cleaning procedure by using acetone and methanol or RCA pre-furnace clean steps to remove any organic and metallic containments. The fabrication starts with a UV photolithography step, where a 500 nm thick LOR-3b at 180 °C for 5 min. and a 5 μm thick AZ12xt are used to establish a lift-off profile for the bottom electrode of the piezoelectric resonator. Lift-off was performed in heated acetone at 60 °C for 30 min, followed by DI water rinsing and N_2_ drying. Photolithography was performed using an EVG620 Mask Aligner (EV Group, St.Florian am Inn, Austria) with standard UV exposure parameters optimized for AZ 12XT photoresist. When employing highly conformal deposition techniques such as RF magnetron sputtering, it is crucial for the thickness of the LOR to be at least 1.5 times greater than the thickness of the deposited metal layer. After the exposure and photoresist development procedures, the wafer undergoes an $O_{2}$ plasma descum process to eliminate any residual photoresist. In this work, Chrome (40 nm) was employed as an adhesive layer for the subsequent metal layer, while Platinum (200 nm) was chosen to form the bottom electrode due to its exceptional chemical resistance and thermal stability, as well as its low electrical resistivity of 10.5 × $10^{-8}$ (Ω∙m). The AJA RF magnetron sputtering tool was used at room temperature, with a chamber pressure of 5 mTorr and a power of 100 W to sequentially deposit a 40 nm thick chromium layer and a 200 nm thick platinum layer, as shown in Figure 1b. Thereafter, the RF magnetron reactive sputtering process was carried out to deposit a 700 nm thick ZnO layer at 100 W power and 5 mTorr chamber pressure that was meticulously executed using optimal parameters to establish a crystal orientation aligned along the (002) c-axis, as shown in Figure 1c. Then, via holes were patterned to establish ohmic contact between the top and bottom electrodes, followed by immersion of the wafer in a ZnO wet etch solution containing 1 HCL:100 $H_{2}O$ to etch away ZnO at the via-hole locations. Following that, the top electrode, composed of a 40 nm layer of chromium (Cr) and a 200 nm layer of platinum (Pt), was then created by using photolithography and the lift-off technique, as shown in Figure 1d. Metal deposition was performed using an AJA International ATC-2200 RF and DC Magnetron Sputtering System (AJA International, Inc., Hingham, MA, USA), applying sequential RF sputtering for ZnO and DC sputtering for Cr/Pt electrodes. Dry etching processes were conducted on an Adixen AMS 100 Deep Reactive Ion Etcher (Alcatel Vacuum Technology, Annecy, France). The backside etch process was employed then for releasing the device. The backside releasing process involves using a 12 μm thick AZ12xt photoresist to define specific cavity releasing areas on the backside of the wafer. The Si handle layer of the SOI wafer was etched and removed by the Bosch high-aspect-ratio (HAR) Si deep reactive ion etching process, as shown in Figure 1e. The buried oxide layer (BOX) was then etched away by using a directional reactive ion etching (RIE) process, as shown in Figure 1f. Next, the process continued from the front side of the SOI wafer. A ZnO reactive ion etching (RIE) process was used to define the resonator body precisely by selectively removing ZnO until the surface of the Si substrate was exposed, as shown in Figure 1g. Then, silicon anisotropic deep reactive ion etching (DRIE) was used to create the body of the resonator by removing material from the pre-released device layer that was already suspended, as shown in Figure 1h. Ultimately, the utilization of low-power $O_{2}$ plasma in a descum process can effectively eliminate remaining photoresist residues prior to device measurement. Both rectangular-plate and disk-shaped resonators were fabricated as depicted in Figure 1. Figure 2 presents a top-view scanning electron microscope (SEM) photo of a fabricated rectangular-plate resonator, highlighting the precision of the fabrication process. The fabricated rectangular-plate resonators consist of a 96 μm × 180 μm rectangular-shaped resonator body with 4 μm wide anchors, while the disk-shaped resonators have a 300 μm diameter along with 4 μm wide anchors.

### 2.2. Device Operation and Simulation Model

The resonators in this work operate by employing a ZnO piezoelectric transducer to actuate the released resonator body to vibrate in a strategically designed resonance mode. The modal analysis simulation of the resonator body is conducted by exploring the state of the released resonator body at its eigen frequency by applying mechanical forces that correspond to the target mode shape of the resonator body structure at its natural resonance frequency. By subjecting the piezoelectric thin-film layer to an electric field between the top and bottom electrodes, it undergoes deformation, thus resulting in a phenomenon known as the piezoelectric effect. When an AC sinusoidal electric field is applied, and its frequency matches the resonator body’s resonance frequency, the released resonator body is stimulated to vibrate in its corresponding resonance mode. A thin-film piezoelectric-on-silicon (TPoS) resonator typically consists of a thin-film piezoelectric transducer layer sandwiched between two metallic electrodes, which are placed on a low-mechanical-loss resonator body composed of a substrate material such as crystalline silicon. The resonance frequency of ZnO-on-Si resonators is primarily determined by the effective acoustic velocity of the resonator body structure, which comprises a stack of thin-film ZnO piezoelectric layers, top and bottom electrode layers, and a Si structural layer. As a result of the resonator body structural materials, piezoelectric micromechanical lateral-extensional resonators could exhibit *Q*-factors ranging from moderate (<1000) to high (>10,000), while resonance frequencies range from tens of MHz to a few GHz, depending on the excited mode. The acoustic velocity of single-crystal silicon is significantly greater than that of sputtered ZnO thin film. Applying an AC actuation signal to the piezoelectric thin-film transducer causes the ZnO piezoelectric transducer layer and the underlying silicon device layer to resonate together. For instance, the lateral extensional mode involves the expansion and contraction of the Si resonator body through the converse piezoelectric effect. The resonator body’s deformation causes periodic piezoelectric charges to be generated throughout the resonator body and detected by the output electrodes in a reciprocal manner. Figure 3 depicts the modal vibration of the 5th lateral extensional mode of the device at the resonance frequency of 191.64 MHz, which is designed with five interdigitated (IDT) top electrodes to vibrate at its 5th-order mode.

In the process of localized annealing investigated in this work, a direct current (DC) is meticulously applied through the resonator body to induce Joule heating, which is essential for resulting in interface modification in the TPoS MEMS resonators. This innovative technique targets the precise elevation of temperature at specific regions of the resonator structure to facilitate metal diffusion and silicide formation, which are crucial for enhancing the device’s mechanical properties. The application of the DC current is carefully controlled in terms of amplitude and duration to ensure that only the designated areas of the chosen resonator, such as the resonator body and anchors, are heated, thus avoiding the potential degradation or structural damage to adjacent components. As depicted in Figure 4, the localized annealing setup for Joule heating has a strategically designed electrode configuration that ensures the DC current flows predominantly through the side-supporting anchors into the resonator body. This DC current generates heat within the resonator body microstructure due to the electrical resistance to achieve the desired modified interface properties without exceeding the material thresholds that could lead to catastrophic failures. The induced heat causes the platinum from the bottom electrode layer to diffuse into the SOI’s silicon device layer, forming a platinum silicide interface. This interface notably alters the physical properties of the resonator body’s stiffness and density, thus resulting in enhanced performance characteristics (i.e., elevated *Q*-factor and reduced motional resistance). The precise control of this localized annealing process in terms of current amplitude and duration ensures that these modifications are localized, preserving the overall integrity and function of the surrounding structures of the MEMS resonator device. It is important to note that the Joule heating mechanism in this study is purely based on direct resistive heating of the resonator structure, without the involvement of plasmonic nanoparticles. The localized Joule heating process was performed by applying a controlled DC current to the resonator electrodes using a source meter. The current was increased gradually until the simulated target temperature (approximately 1060 °C) was reached at the anchor regions, as confirmed by prior calibration with the simulation model. The applied current ranged between 5 mA and 15 mA, depending on the device geometry. The heating duration was typically 30 s to 2 min to ensure sufficient thermal diffusion for interface modification without damaging the resonator structure. The applied voltage and current were continuously monitored to calculate the resistive Joule heating power and avoid thermal overshoot. Figure 5 presents a COMSOL 6.1 Multiphysics simulation of an array of identical rectangular-plate resonators, which indicates the localized annealing of one resonator to reach a temperature up to 1060 °C (primarily at its side-supporting anchors) would not affect the adjacent devices, as they stay below 70 °C. Obviously, this is not achievable by any traditional annealing processes by using a furnace, an oven, or a hot plate.

COMSOL Multiphysics was used to simulate both the modal response (Figure 3) and the thermal behavior during localized Joule heating (Figure 5 and Figure 6 of the ZnO-on-Si MEMS resonators. The full material stack, including the silicon device layer (10 µm), ZnO layer (700 nm), and Cr/Pt top and bottom electrodes (40/200 nm), was modeled to reflect real device geometry. In the modal analysis (Figure 3), fixed boundary conditions were applied at the side-supporting anchors, while the rest of the resonator body surfaces were left unconstrained. A mechanical harmonic excitation was applied to simulate lateral extensional mode vibration, and eigenfrequency analysis was performed to extract mode shapes and resonant frequencies. For thermal simulations (Figure 5 and Figure 6), resistive Joule heating was modeled by applying a current density across the anchors using the electric currents module. A stationary thermal analysis was used to evaluate the steady-state temperature distribution. Adiabatic (thermally insulated) boundary conditions were assumed for all external faces, except the heat-generating anchor regions. A fine tetrahedral mesh with localized refinement was used to ensure convergence near interfaces and anchors. Material properties (e.g., thermal conductivity, density, specific heat, and electrical resistivity) were defined based on experimentally validated values reported in the prior work literature for ZnO, Si, Cr, and Pt. Simulation results confirmed that temperature elevation occurred predominantly at the anchors, validating the localized nature of the annealing. This model also confirmed that adjacent resonators remain unaffected due to thermal isolation characteristics of the silicon-on-insulator (SOI) wafer substrate (Figure 5). 

Figure 6 depicts a COMSOL Multiphysics thermal simulation of a disk-shaped resonator and rectangular-plate resonator, which both have side-supporting anchors. In the COMSOL simulation, the effect of the localized annealing is shown under conditions identical to those applied during the experimental study conducted in this work. In addition to the thermal distribution analysis, we examined the thermal stresses induced by the localized Joule heating using COMSOL 6.1 Multiphysics simulations as shown in Figure 6. The results indicated that the highest thermal-induced stresses were localized at the anchor regions, which correspond to the areas of maximum temperature due to resistive heating. Importantly, the simulated maximum stress levels remained well below the fracture strength and yield limits of both the silicon and ZnO layers, ensuring the structural integrity of the resonator during and after the annealing process. These findings confirm that the localized annealing method does not introduce mechanical failure risks and support its suitability as a post-fabrication enhancement technique.

## 3. Results and Discussion

An approximation model can be used for the resonance behavior of a micromachined resonator at its resonance mode. For the ZnO-on-Si resonator composed of vertically stacked structural layers, the equivalent acoustic velocity $v_{eq}$ can be defined as [25]:(1)$$v_{eq}=\sqrt{\frac{E_{1}T_{1}+E_{2}T_{2}+\ldots +E_{n}T_{n}}{\left(\rho_{1}T_{1}+\rho_{2}T_{2}+\ldots +\rho_{n}T_{n}\right)\left(1-\sigma_{n}^{2}\right)}}$$
where $E_{n},\;\rho_{n},\;\sigma_{n},\;\mathrm{and}\;T_{n}$ are the respective Young’s modulus, density, Poisson’s ratio, and thickness of each constituent layer of the stacked structural materials, including the top/bottom metal electrode layer, ZnO piezoelectric transducer layer, and silicon device layer of the SOI substrate. An estimation of the resonance frequency can be obtained for both the rectangular-plate and the disk-shaped resonators [26,27]:(2)$$f_{rec}=\frac{1}{2\left(W/N\right)}\sqrt{\frac{E_{eq}}{\rho_{eq}}}=\frac{1}{2L_{p}}v_{eq}$$(3)$$f_{disk}=\frac{\alpha }{R}v_{eq}$$
where W is the width of the suspended rectangular-plate resonator body, $E_{eq}\;$ is the equivalent Young’s modulus of the stacked and suspended resonator body, *N* is the number of the interdigitated (IDT) electrode fingers, and $L_{p}$ is the pitch size. The dimensions and layout of the rectangular-plate resonator are clearly depicted in its top-view diagram, as shown in Figure 7, illustrating its configuration. Figure 8 shows a top-view schematic of the disk resonator, depicting its key dimensions and geometric arrangement. *R* is the radius of the disk resonator, and α is the mode-dependent frequency constant. The resonator can be modeled as simple as a series RLC (*R*_*m*_, *L*_*m*_, *C*_*m*_) circuit, where *R*_*m*_, *L*_*m*_, and *C_m_* represent the motional resistance, motional inductance, and motional capacitance, respectively. The motional resistance of a rectangular-plate resonator operating in its fundamental mode can be expressed as [26,28]:(4)$$R_{m}\approx \frac{\pi t_{tot}\sqrt{E_{eq}\rho_{eq}}}{2d_{31}^{2}E_{ZnO}^{2}WQ}$$
where $d_{31}$ is the transverse piezoelectric coefficient, $E_{ZnO}$ is the Young’s modulus of the ZnO layer, and *Q* is the quality factor of the resonator. Figure 9 presents a conceptual diagram that correlates localized annealing by Joule heating with induced changes in the material properties of the structural materials of the resonator body and anchors [29]. As a result, the Young’s modulus of both the resonator body and its side-supporting anchor is increased after localized annealing that leads to a slight resonance frequency increase and greatly increased energy stored per cycle, along with reduced anchor-related losses through those annealed side-supporting anchors. As shown in Figure 5, the side-supporting anchors could reach a temperature as high as 1060 °C, as the highest temperature of the entire resonator body for localized annealing by Joule heating.

Increasing the temperature by localized annealing enables the formation of the ${Pt}_{2}Si$ and PtSi platinum silicide. First, Pt diffuses to single-crystal Silicon to form ${Pt}_{2}Si$ around 180–200 °C. Then, if the temperature is maintained increasing to reach 280–400 °C, then Si atoms started to convert from ${Pt}_{2}Si$ to form PtSi that ultimately increases the overall Young’s modulus [30].

As illustrated in Figure 10 and Figure 11, localized annealing causes Pt to diffuse into the Si resonator body layer. An enhancement in Young’s modulus through the formation of the silicide in the vibrating body also leads to a modification in the equivalent Young’s modulus, designated as $E_{eq}$. This compositional modification results in a slight alteration in frequency and *Q* while leading to a decrease in the motional resistance. The necessity arises naturally for further in-depth investigation of the interface layer to intellectualize our approach and to support our hypothesis on silicide formation. For this purpose, transmission electron microscopic (TEM) samples, both before and after annealing conditions, were prepared with a focused ion beam (FIB) tool. Thereafter, 100 nm thick specimens before and after localized annealing were investigated by a Tecnai F20 transmission electron microscopy (TEM) (Field Electron and Ion Company, Hillsboro, OR, USA). The TEM metrology analysis results clearly demonstrate the effect of interface modification through localized annealing will be discussed later. The formation of PtSi at the interface was confirmed by Energy Dispersive Spectroscopy (EDS) analysis, with the corresponding reference pattern presented in Table 1.

For the purpose of improving the overall resonator performance, several researchers have investigated different approaches; some are based on Joule heating [8,9,10,11,12,13,14,15]. However, these previously studied resonators mostly operate with different transduction mechanisms, such as capacitive and piezoresistive, with only the exception of the work by Wei et al., where the transduction mechanism is also piezoelectric [15]. Moreover, our work herein differentiates from this prior work by employing single-crystal silicon instead of plated nickel as the main structural material of a vibrating resonator body. Although silicide formation for the semiconductor device research has been investigated for more than 40 years, this is one of the first studies in the literature that systematically investigates and correlates the silicide formation with the ZnO piezoelectric-on-silicon MEMS resonators’ performance to the best of our knowledge. More importantly, this work demonstrates that the *Q*-factor of the resonator was increased by over 300% through the localized annealing process under optimal conditions, while the motional impedance was lowered accordingly.

The electrical characterization of the MEMS resonators was performed using a Keysight E5061B Vector Network Analyzer (VNA) (Keysight Technologies, Santa Rosa, CA, USA) connected through Cascade Microtech GSG RF probes on a probe station. All RF probe measurements were conducted in ambient air at room temperature. The full S-parameter frequency responses of the resonators were measured, and key parameters such as the resonance frequency, motional resistance, and *Q*-factor were extracted using modified Butterworth–Van Dyke (mBVD) equivalent circuit fitting. Figure 12 shows the measured frequency responses for a rectangular-plate resonator, showcasing improvements in key performance metrics such as the *Q*-factor and motional resistance after the localized annealing process. Figure 13 shows the measured frequency responses of a disk resonator, which also demonstrates the enhancements in performance after the localized annealing treatment. Localized annealing enables the designer to anneal the device selectively in case frequency trimming is needed while also offering an effective means to enhance the overall resonator performance further.

The annealing process is initiated by the precise application of a direct current (DC), which selectively heats the resonator body through predetermined side-supporting anchors. This localized Joule heating approach is critical, as it ensures the platinum diffusion occurs only at specific regions within the resonator body and anchors, thus maintaining the integrity and functionality of the annealed resonator device while not thermally impacting any of the adjacent devices situated on the same wafer substrate. The temperature is carefully ramped to specific thresholds to control the formation of different platinum silicide phases, each of which could impact the resonator’s properties differently. At the onset, around 180 °C, the first phase of platinum silicide, Pt_2_Si, begins to form. This Pt_2_Si phase is crucial, as it lays the foundational microstructure for further silicide development. As the annealing process continues and the temperature reaches approximately 260 °C, this phase transitions predominantly into Pt_2_Si. This phase is known for its potential to enhance the structural stability and uniformity of the silicide layer, which is beneficial for the mechanical properties of the resonator. The process is not complete until the temperature escalates to 360 °C, at which temperature the PtSi phase starts to form [31]. A cross-sectional sample was prepared by milling the device using a Quanta 200 3D Dual Beam Focused Ion Beam (FIB) system (Field Electron and Ion Company, Hillsboro, OR, USA). The sample was thinned down to approximately 100 nm to enable high-resolution TEM imaging and interface analysis, as shown in Figure 14. This final phase is noted for its superior mechanical properties, which are essential for achieving enhancements in Young’s modulus and overall performance.

Figure 15 and Figure 16 present TEM photos indicating the formation of PtSi of (101) and (021) orientation, with characteristic lattice constants of 3.08 Armstrong and 2.2 Armstrong, respectively. Also, Figure 17 depicts a comparison of Energy Dispersive Spectroscopy (EDS) analysis results from annealed and unannealed TEM samples, which clearly show that the locally annealed sample has a substantial amount of Pt diffused into the Si device layer up to 1 micrometer away from the original Pt/Si interface. 

The experimental results showed a clear improvement in the *Q*-factor and a reduction in the motional resistance after localized Joule heating, in agreement with the trends predicted by the simulation. The thermal simulations confirmed that the highest temperature occurs at the anchor regions, which correlates with the observed reduction in anchor-related energy dissipation and the improvement in the resonator performance. Furthermore, the modal analysis simulations predicted resonance frequencies that closely matched the experimentally measured values, confirming the accuracy of the device modeling. These findings validate that the simulation models reliably capture the thermal, mechanical, and acoustic behavior of the resonators, supporting the experimental observations and confirming the effectiveness of the localized annealing process.

Table 2 presents a summary of the key improvements in terms of the performance of MEMS resonators facilitated by the localized annealing process. Table 2 shows a comparison of the performance metrics before and after the localized annealing, revealing notable performance enhancement effects for both the rectangular-plate and disk-shaped resonators. Measurements of the as-fabricated rectangular-plate resonator at 192.47 MHz exhibited a *Q*-factor of 916, along with an insertion loss of −29.02 dB. The localized annealing process, which involves the application of a DC current of 0.5 A for 30 min under an average power of 2 W and an energy dosage of 1.5 kJ, resulted in a remarkable performance improvement. As shown, the *Q*-factor was increased to 3632, and the insertion loss was decreased to −14.78 dB, after the localized annealing conducted under the aforementioned conditions. These changes indicate about 4X improvement in terms of the measured *Q*-factor while lowering the insertion loss by more than 14 dB, which underscores the effectiveness of the annealing process. Similarly, the disk-shaped resonator measured before annealing exhibited an initial *Q*-factor of 1590 and an insertion loss of −31.10 dB. After annealing under slightly higher power of 2.5 W for 25 min with an energy dosage of 3.75 kJ, the *Q*-factor was elevated to 2448, and the insertion loss was improved to −21.31 dB. Although these improvements in this disk resonator are slightly less pronounced than those of the rectangular-plate resonator, they still signify substantial performance enhancements for this post-fabrication treatment technique. The measurements before and after localized annealing underscore its critical role in adjusting the material properties of the resonators, notably, through the formation of platinum silicide interfaces that significantly enhance Young’s modulus. The increases in the *Q*-factor indicate an enhanced ability of the resonators to store more energy per cycle after the annealing [15,32], which is also a direct outcome of improved material properties. Meanwhile, since the side-supporting anchors were annealed under the highest temperature when conducting the localized annealing based on the setup as shown in Figure 4, the annealed anchors lower the energy dissipations that are otherwise predominantly ascribed to anchor-related losses. Moreover, the reduction in the insertion loss suggests a more efficient energy conversion of the piezoelectrically transduced resonator, which is crucial for applications requiring high energy efficiency and signal integrity. This study demonstrates that localized annealing through Joule heating is a highly effective method for enhancing the performance parameters of MEMS resonators. The meticulous tweaking of the annealing conditions and the resulting performance enhancements in Table 2 show a clear and detailed correlation between the annealing process parameters and the observed improvements in the resonator performance, paving the way for future innovations.

While the localized annealing technique by Joule heating demonstrated significant potential to improve the *Q*-factor and motional resistance, it has certain limitations. The process requires precise control of the current amplitude and duration to avoid excessive heating, which could potentially damage the resonator or adjacent structures. Additionally, variability in material properties across wafers may affect the uniformity and consistency of the annealing results. The long-term thermal stability of the modified interface and its impact on device aging or reliability under continuous operation also require further investigation. Finally, the applicability of this method may be limited for resonators with geometries that do not allow efficient current flow through the anchor or other regions.

## 4. Conclusions

To assess the effects of localized annealing through Joule heating on the performance of piezoelectric ZnO-on-Si rectangular-plate and disk-shaped resonators, rigorous frequency response characterization before and after the localized annealing process was conducted. The localized annealing process, which employs Joule heating, played an instrumental role in modifying the interface between the bottom electrode and the silicon resonator body through PtSi silicide formation, leading to improved key performance metrics such as an enhanced *Q*-factor and reduced motional resistance. This preliminary study focuses on the investigation of the effect of the Joule heating controlled by localized annealing on the performance of two piezoelectric resonator designs through the formation of PtSi silicide. A substantial improvement in device performance, with a more than 300% increase in the *Q*-factor and significant reductions in the insertion loss and motional resistance, was achieved in rectangular-pate resonators. This highlights the potential of localized annealing to refine the structural properties of MEMS resonators. These advancements are critical for the development of more reliable and energy-efficient MEMS resonator devices, catering to a wide range of real-world device applications, from high-precision frequency and timing control to high-resolution sensing and efficient energy harvesting. Moreover, this work lays a solid foundation for future research and development on this technique, suggesting that considerable enhancements in resonator performance could be achievable through meticulous control of annealing conditions. The correlation of the resonator device-level improvements with changes at the microstructural level, specifically, through silicide formation, provides valuable insights into the material science aspects of MEMS device engineering. This not only fills an important knowledge gap but also expands the potential application spectrum of MEMS resonators for performance-demanding scenarios. By creating innovative fabrication and annealing techniques, this work contributes to advancing the area of microsystem technology, promising greatly enhanced functionalities and performance reliability in future applications.

## Figures and Tables

**Figure 1 micromachines-16-00885-f001:**
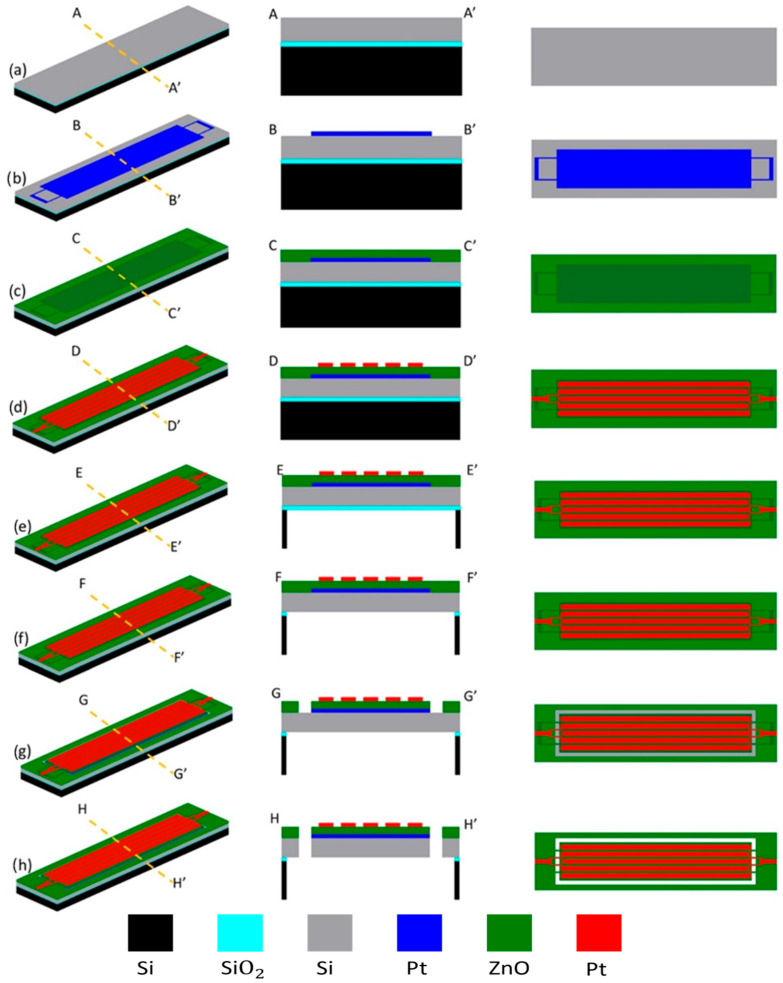
(**a**–**h**) Illustration of the fabrication process flow by depicting key steps of the fabrication in perspective, cross-section, and top-view diagrams.

**Figure 2 micromachines-16-00885-f002:**
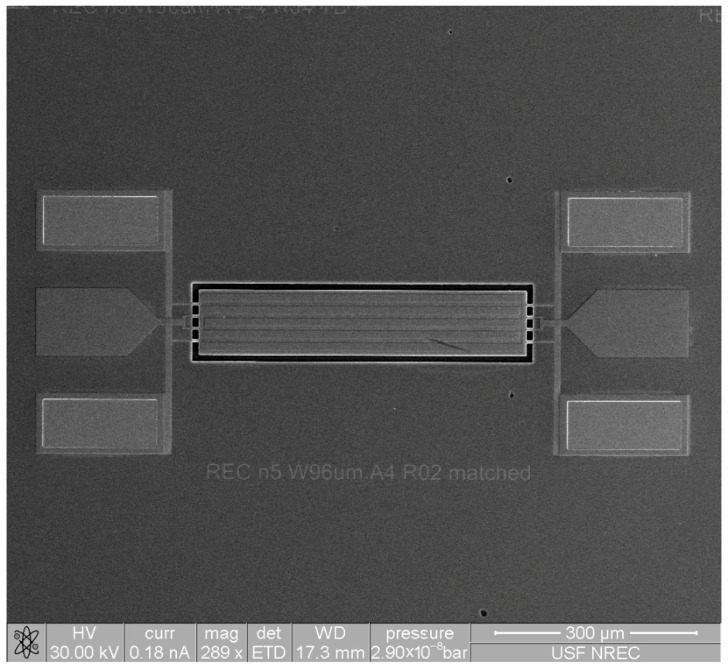
SEM image of a microfabricated, rectangular-shaped ZnO-on-Si resonator.

**Figure 3 micromachines-16-00885-f003:**
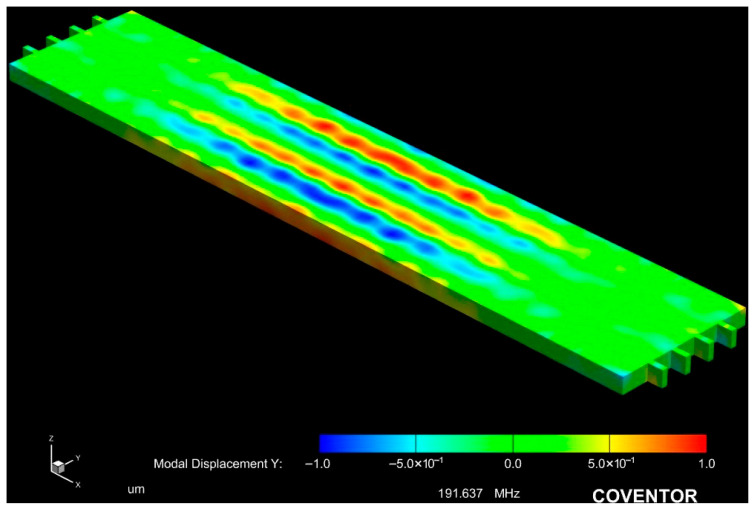
Modal displacement of a rectangular-plate resonator at its 5th-order mode.

**Figure 4 micromachines-16-00885-f004:**
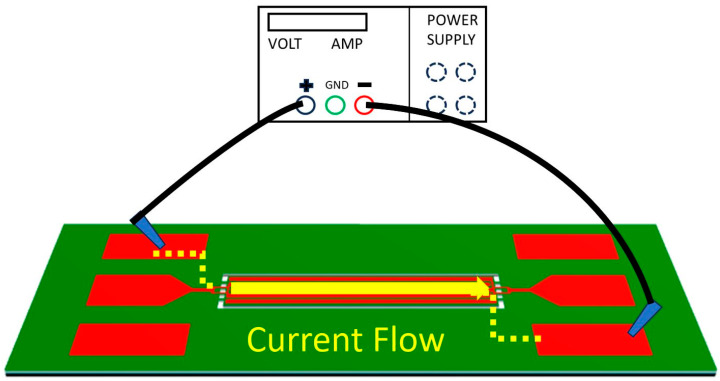
Conceptual illustration of the localized annealing experimental setup, where the strategically chosen amplitude and duration of a DC current is applied to heat the resonator body through its side-supporting anchors and electrodes.

**Figure 5 micromachines-16-00885-f005:**
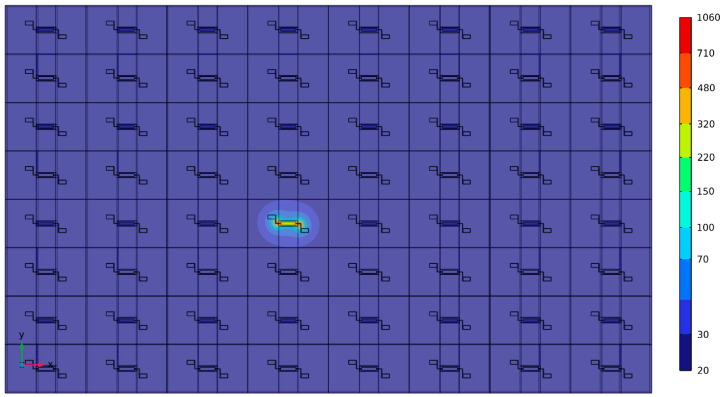
COMSOL Multiphysics thermal simulation of an array of identical rectangular-plate ZnO-on-Si resonator devices under localized Joule heating. The simulation shows that the temperature elevation—indicated by the color bar (in °C)—is confined to the annealed resonator, while adjacent devices remain at significantly lower temperatures. This demonstrates that localized annealing by resistive Joule heating effectively limits the thermal impact on neighboring structures, a result not achievable by conventional annealing methods such as furnace, oven, or hot-plate treatments.

**Figure 6 micromachines-16-00885-f006:**
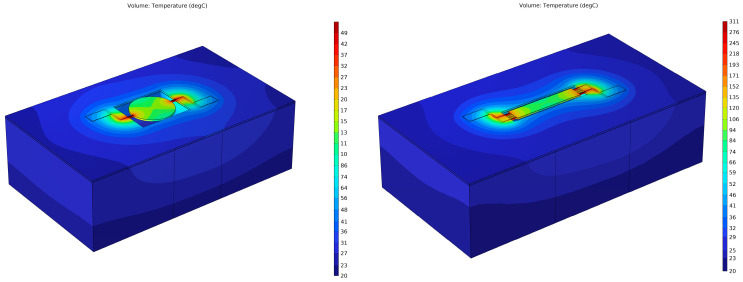
COMSOL Multiphysics thermal simulation of a disk-shaped and a rectangular-plate resonator, both with side-supporting anchors, through localized annealing under conditions identical to those applied during the experimental study conducted in this work.

**Figure 7 micromachines-16-00885-f007:**
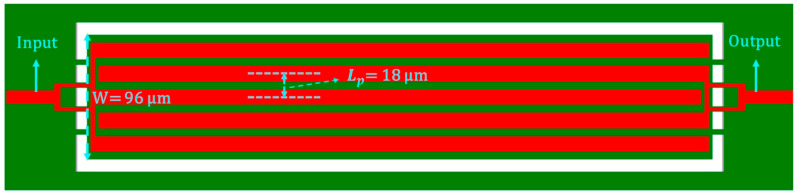
Top-view diagram of a rectangular-plate resonator to illustrate its key dimensions, as well as the layout of its input/output IDT electrodes (shown in red).

**Figure 8 micromachines-16-00885-f008:**
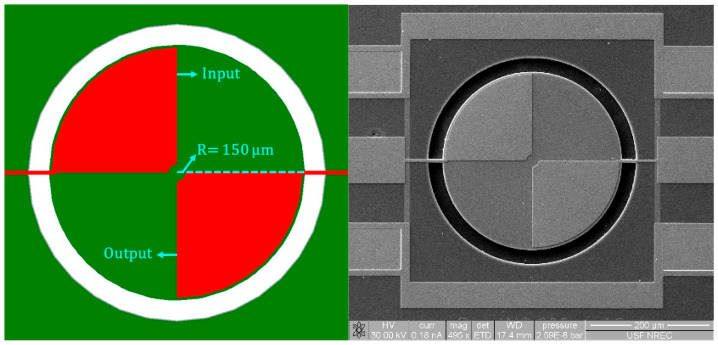
Top-view schematic and SEM photo of a 150 μm radius disk resonator to show its key dimensions and the layout of its two-port input and output electrode configurations.

**Figure 9 micromachines-16-00885-f009:**
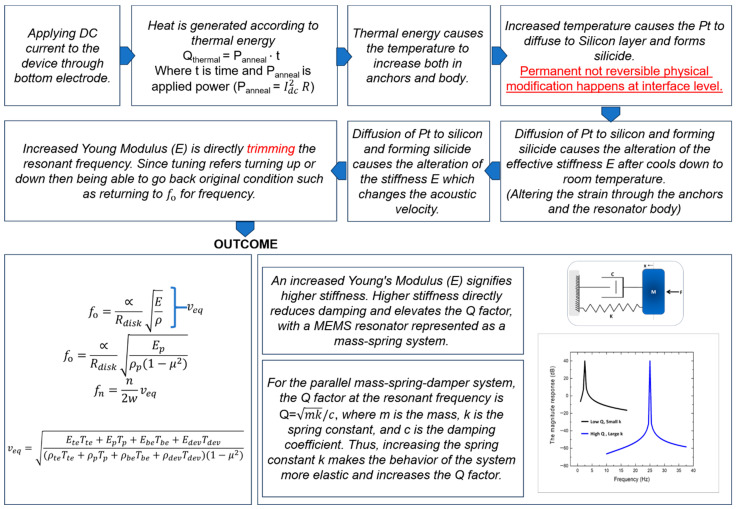
Simplified diagram to showcase the mechanism and effect of localized annealing [29].

**Figure 10 micromachines-16-00885-f010:**
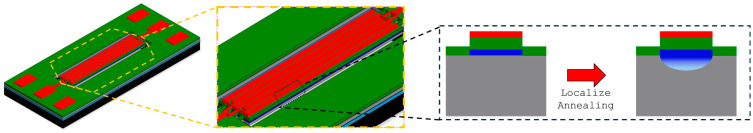
Conceptual illustration of silicide formation through metal diffusion.

**Figure 11 micromachines-16-00885-f011:**
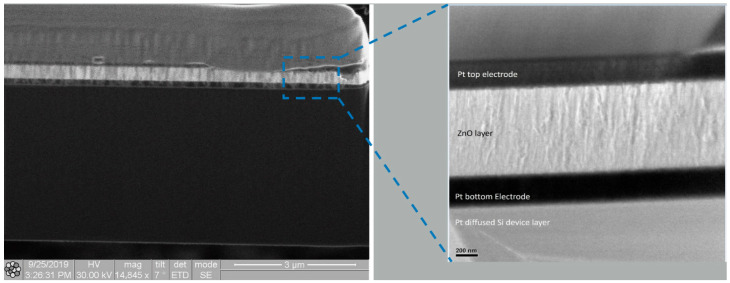
TEM cross-sectional images after localized annealing indicate silicide formation.

**Figure 12 micromachines-16-00885-f012:**
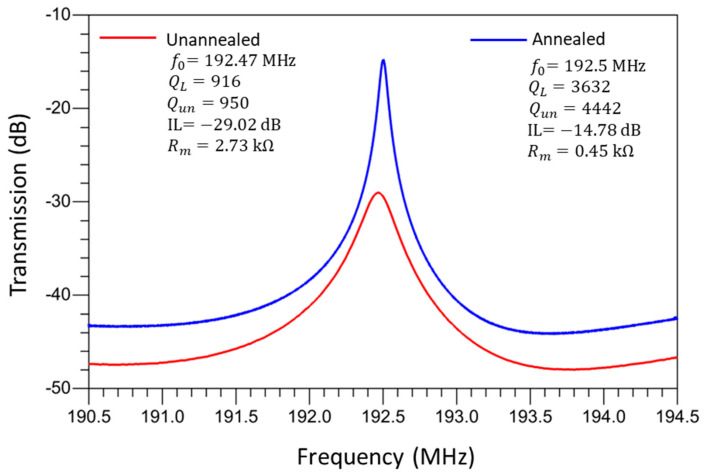
A comparison of measured frequency responses of a rectangular-plate ZnO-on-Si piezoelectric resonator before and after localized annealing.

**Figure 13 micromachines-16-00885-f013:**
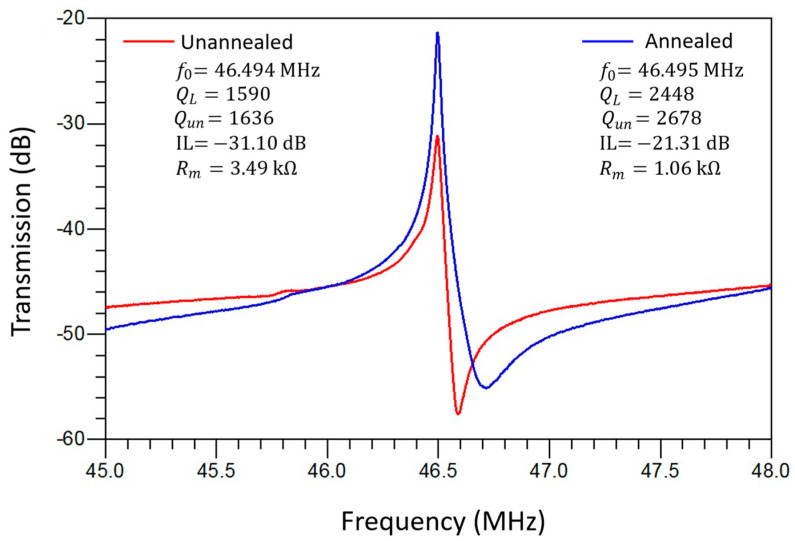
A comparison of measured frequency responses of a disk-shaped ZnO-on-Si piezoelectric resonator before and after localized annealing.

**Figure 14 micromachines-16-00885-f014:**
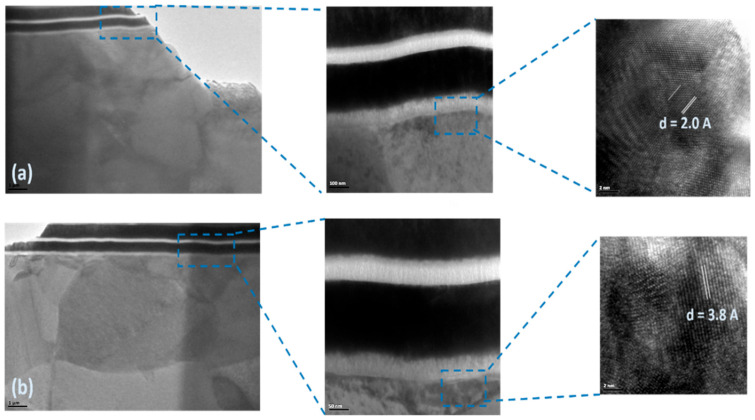
TEM images for a FIB-prepared cross-sectional specimen from a rectangular-plate resonator by showing resonator layer interfaces: (**a**) before annealing and (**b**) after annealing.

**Figure 15 micromachines-16-00885-f015:**
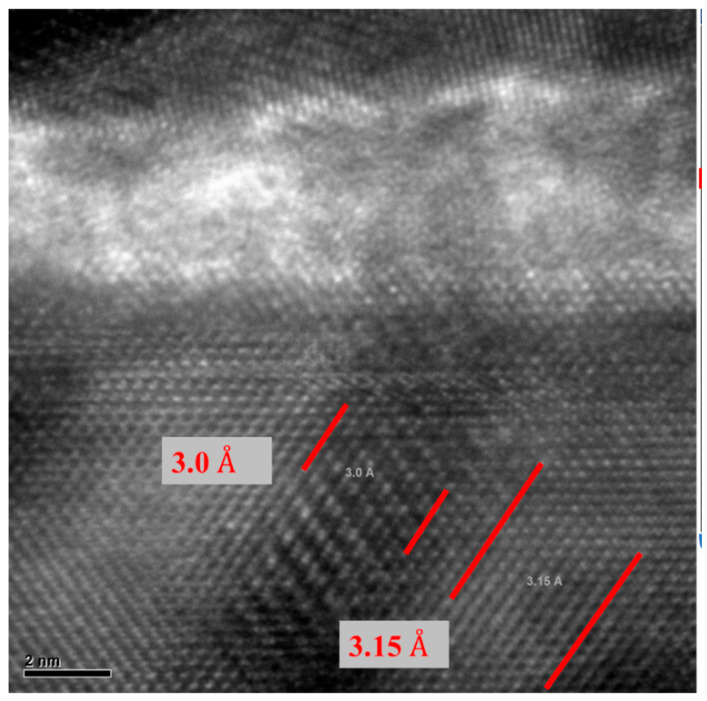
TEM image of snowplow-like regions where PtSi formations were observed. The presence of PtSi was confirmed through Energy Dispersive Spectroscopy (EDS) analysis using the X’Pert HighScore software (version 1.0f). The corresponding reference pattern is presented in Table 1.

**Figure 16 micromachines-16-00885-f016:**
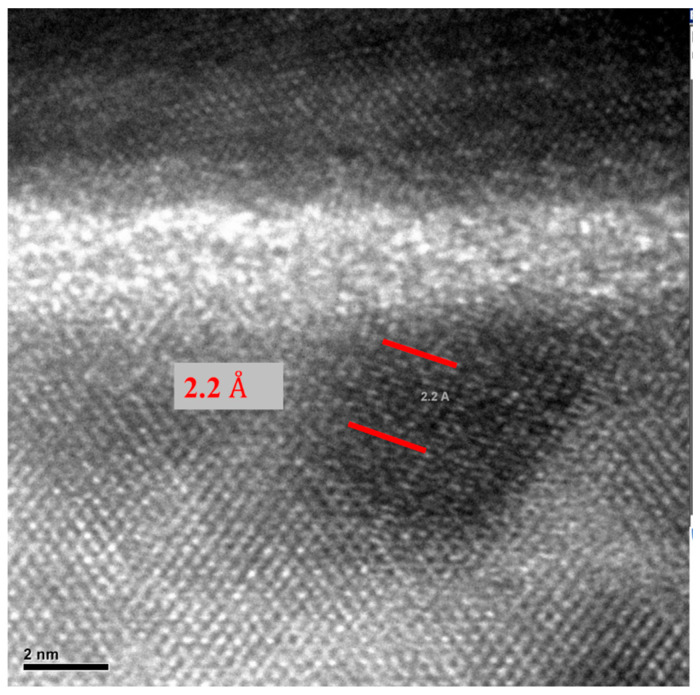
TEM image of snowplow-like regions where PtSi formations were observed. The presence of PtSi was confirmed by Energy Dispersive Spectroscopy (EDS) analysis using the X’Pert HighScore software. The corresponding PtSi reference pattern is presented in Table 1.

**Figure 17 micromachines-16-00885-f017:**
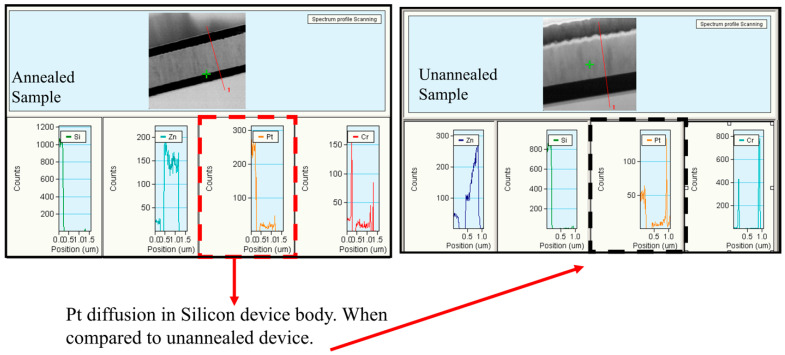
A comparison of Energy Dispersive Spectroscopy (EDS) analysis results from annealed and unannealed TEM samples, which clearly show that the localized annealed sample has a substantial amount of Pt diffused into the SOI’s Si device layer away from the original Pt/Si interface.

**Table 1 micromachines-16-00885-t001:** Reference pattern of PtSi obtained using the X’Pert HighScore EDS analysis software.

Peak List				
No.	h	k	l	d (A)
1	1	1	0	4.070
2	1	0	1	3.080
3	2	0	0	2.970
4	0	2	0	2.896
5	0	2	1	2.216
6	2	1	1	2.122
7	1	2	1	2.075
8	2	2	0	2.037
9	3	1	0	1.867
10	0	0	2	1.804

**Table 2 micromachines-16-00885-t002:** Measured results and annealing conditions for both types of resonators.

Process Step Name	Current (A)	Average Power (W)	Time (min)	Energy Dosage (kJ)	Resonant Frequency (MHz)	*Q*-Factor	Insertion Loss (dB)
Unannealed Rectangular	-	-	-	-	192.468	916	−29.02
Annealed Rectangular	0.5	2	30	1.5	192.500	3632	−14.78
Unannealed Disk	-	-	-	-	46.494	1590	−31.10
Annealed Disk	0.5	2.5	25	3.75	46.495	2448	−21.31

## Data Availability

Measurements and other relevant data are available from the corresponding authors upon request via email.

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
