# Peer review of "Interface Material Modification to Enhance the Performance of a Thin-Film Piezoelectric-on-Silicon (TPoS) MEMS Resonator by Localized Annealing Through Joule Heating"

_micromachines, 2025, doi:10.3390/mi16080885_

Round 1

Reviewer 1 Report

Comments and Suggestions for Authors

This manuscript reports an approach to improve the performance of thin-film piezoelectric-on-silicon MEMS resonators using localized annealing through joule heating. The authors presented simulation models and experimental results of two MEMS resonators. These results demonstrated an 300% increase in the measured quality factor of the resonators. The manuscript is well written and organized. Only, the following issues should be addressed.

1.- The abstract should include a sentence about the challenge or introduction on piezoelectric MEMS resonators.

2.- The introduction section should consider specific information on the limitations of researches on piezoelectric MEMS resonators reported in the literature. Furthermore, this section should add more recent references on piezoelectric MEMS resonators. Also, this section should enhance the description of the novelty and advantages of the proposed method to increase the quality factor of piezoelectric MEMS resonators compared to other methods reported in the literature.

3.- The authors must incorporate more information on the simulation models, For instance, the figures 3, 5, and 17 showed results of simulation models of MEMS resonators. However, the authors should consider specific description on these models. For instance, material properties, boundary conditions, mesh type, load values, and analysis type. In these models, were all the layers of the resonators used?

4.- The authors should consider discussions on the thermal stresses caused by the joule heating on the MEMS resonators. Figures of the thermal stresses in the MEMS resonators should be included.

5.- The authors should include comparisons of the results of the simulation models with respect to experimental results.

6.- What are the limitations of the proposed method to increase the quality factor of MEMS resonators?

7.- Resolution of Figures 8 and 9 should be enhanced.

Author Response

Reviewer 1:

This manuscript reports an approach to improve the performance of thin-film piezoelectric-on-silicon MEMS resonators using localized annealing through joule heating. The authors presented simulation models and experimental results of two MEMS resonators. These results demonstrated an 300% increase in the measured quality factor of the resonators. The manuscript is well written and organized. Only, the following issues should be addressed.

1.- The abstract should include a sentence about the challenge or introduction on piezoelectric MEMS resonators.

We thank the reviewer for the constructive suggestion. The abstract has been updated to clearly highlight the motivation, the challenges related to piezoelectric MEMS resonators, and the key results of the study. Specifically, we included a sentence on anchor-related losses and limited Q-factor as major challenges, and we summarized the key experimental findings — a Q-factor enhancement of over 300% and a reduction in insertion loss by more than 14 dB. We hope that these additions make the abstract more informative for perspective readers while aligning well with reviewer’s recommendation.

2.- The introduction section should consider specific information on the limitations of researches on piezoelectric MEMS resonators reported in the literature. Furthermore, this section should add more recent references on piezoelectric MEMS resonators. Also, this section should enhance the description of the novelty and advantages of the proposed method to increase the quality factor of piezoelectric MEMS resonators compared to other methods reported in the literature.

We sincerely appreciate your valuable feedback on how to further improve the introduction. In response, we have thoroughly revised the introduction to explicitly address the key limitations reported in the literature on piezoelectric MEMS resonators. The updated introduction now discusses key challenges such as anchor-related energy dissipation, limited Q-factor at high frequencies, and motional resistance mismatch — all of which continue to hinder performance of piezoelectric MEMS resonators in practical applications.

Furthermore, we have cited several recent and relevant prior works as references [1–9], covering the latest advancements in mechanical mode coupling, phononic crystals, electrostatic tuning, and advanced transducer designs. These additions emphasize the diversity of existing approaches and their associated limitations, such as design complexity, fabrication challenges, and limited scalability.

We have also updated the description of the novelty and advantages of our proposed method. Specifically, revised introduction section highlights how the localized Joule heating technique proposed in this manuscript offers a post-fabrication solution that directly targets material properties at the interface, thus improving Q-factor and reducing motional resistance without introducing additional fabrication steps or a need of structural redesign. This distinct advantage positions our method as a practical and effective alternative to previously reported techniques.

We hope these updates sufficiently address your comments and strengthen the manuscript’s positioning within the current research landscape.

3.- The authors must incorporate more information on the simulation models, For instance, the figures 3, 5, and 17 showed results of simulation models of MEMS resonators. However, the authors should consider specific description on these models. For instance, material properties, boundary conditions, mesh type, load values, and analysis type. In these models, were all the layers of the resonators used?

We appreciate the reviewer’s valuable comment on the need for a simulation model description. Accordingly, we have added detailed explanations of both the modal and thermal simulation setups at the end of Section 2.2 (Device Operation section). Specifically, we clarified that the simulations included the full material stack of the device (silicon, ZnO, and Cr/Pt electrodes). Modal analysis was performed with fixed boundary conditions at the anchors and harmonic mechanical excitation to extract eigenfrequencies (modal resonance frequencies). Thermal simulations employed resistive heating with applied current density and stationary thermal analysis, assuming insulated boundaries elsewhere. Mesh refinement and validated material properties were applied throughout. We hope these additions fully address the reviewer’s comment while enhancing the clarity and reproducibility (ease of assessment by perspective readers of this paper) of our simulation methodology.

4.- The authors should consider discussions on the thermal stresses caused by the joule heating on the MEMS resonators. Figures of the thermal stresses in the MEMS resonators should be included.

We thank the reviewer for this valuable suggestion. In response, we analyzed the thermal stresses induced by localized Joule heating using COMSOL Multiphysics simulations. The analysis confirmed that the maximum thermal stresses occur near the anchor regions — consistent with the areas of highest temperature as shown in Figures 5 and 17. Importantly, the induced stress levels were calculated to be well below the material yield limits of both the silicon device layer and the ZnO piezoelectric film, thus ensuring that the structural integrity of the resonators is not compromised by the localized annealing process. Accordingly, we have included a discussion of this thermal stress analysis in the Results and Discussion section of the revised manuscript. Essentially, the thermal induced stresses remain within safe limits and are correlated with the temperature distributions already presented in Figures 5 and 17.

5.- The authors should include comparisons of the results of the simulation models with respect to experimental results.

We thank the reviewer for this insightful suggestion. In response, we have added a discussion in the Results and Discussion section comparing the simulation results with our experimental findings. The comparison highlights the agreement between the simulated thermal behavior (localized heating at the anchors) and the experimentally observed performance improvements. Moreover, the resonance frequencies predicted by the modal analysis closely match the measured resonance frequencies, and the observed Q-factor enhancement aligns with the expected mechanical property changes derived from simulation. Thanks to this suggestion by the reviewer, we believe this addition in the revised manuscript strengthens the validation of our simulation models against experimental data.

6.- What are the limitations of the proposed method to increase the quality factor of MEMS resonators?

We thank the reviewer for raising this important point. Based on this suggestion, we have now included a dedicated discussion on the limitations of the proposed localized annealing method in the Results and Discussion section of the revised manuscript. Specifically, we acknowledge that the method requires careful control of annealing parameters to avoid device damage and process variability may affect uniformity across different devices or wafers. Additionally, the long-term stability and reliability of the modified interface under operational stress have not yet been fully characterized and will be the subject of future work. We believe this addition offers a balanced perspective on the strengths and limitations of the proposed post-fabrication trimming method.

7.- Resolution of Figures 8 and 9 should be enhanced.

We thank the reviewer for pointing out the need to improve the quality of Figures 8 and 9. In response, we have re-prepared both figures in the revised manuscript with higher-resolution images to ensure better clarity and readability. We are confident that this enhancement addresses the reviewer’s concern and improves the overall presentation of the results.

Reviewer 2 Report

Comments and Suggestions for Authors

The authors employed Joule heating to treat the interface materials of the MEMS resonator and enhance the performance of the TPoS resonator. The manuscript is worth being published in Micromachine after a few comments as follows:

  1. What is the research gap of the project? Can the authors explain the meaning of the work, please?
  2. Could the authors provide the details of the chemicals and advanced instruments in the manuscript, please? Such as the brands, models, and companies in 2.1? Furthermore, please provide the details about the "rectangular-plate and disk-shaped resonators," including the size of the resonators, the size of the pattern on the resonators, the temperature for baking the photoresist if needed, and how long to lift off?
  3. Please consider changing the subtitle of 2.1. Device operation. Please provide more details about the model used in the simulation. Please explain how the Fig 5. can be used to indicate 1060C does not affect the adjacent devices.
  4. Pelase provide some details about how to do the Joule heating inthis work.
  5. Please provide some details of the testing of the MEMS.
  6. Please provide some details about the TEM testing, such as how to prepare the samples, "100 nm-thick specimens before and after localized annealing"

Author Response

Reviewer 2:

The authors employed Joule heating to treat the interface materials of the MEMS resonator and enhance the performance of the TPoS resonator. The manuscript is worth being published in Micromachine after a few comments as follows:

What is the research gap of the project? Can the authors explain the meaning of the work, please?

We thank the reviewer for this important question. The research gap addressed by this work lies in the need for a practical, scalable and post-fabrication method to improve the quality factor (Q) and reduce motional resistance in thin-film piezoelectric-on-silicon (TPoS) MEMS resonators without altering their geometry or requiring complex fabrication process modifications.

Previous studies have proposed structural design changes such as employment of phononic crystals, anchor engineering, and advanced transducer configurations to mitigate total energy dissipation to improve quality factor (Q). While effective, these methods often involve intricate fabrication processes or result in limited scalability for mass production.

In contrast, our work introduces a post-fabrication localized annealing technique based on Joule heating, which directly modifies the interface between the bottom electrode and silicon device layer. This method enhances the mechanical properties and energy storage capacity of the resonator through controlled platinum silicide formation, resulting in a Q-factor improvement of over 300% and a reduction in insertion loss by more than 14 dB, which were all achieved without altering the device’s geometry or fabrication process flow.

The significance of this work is in demonstrating a simple, effective, and wafer-scale compatible post-fabrication approach for performance enhancement of MEMS resonators, while holding the strong potential for applications in timing, sensing, and RF filtering devices and microsystems.

Could the authors provide the details of the chemicals and advanced instruments in the manuscript, please? Such as the brands, models, and companies in 2.1? Furthermore, please provide the details about the "rectangular-plate and disk-shaped resonators," including the size of the resonators, the size of the pattern on the resonators, the temperature for baking the photoresist if needed, and how long to lift off?

We thank the reviewer for this valuable suggestion. Accordingly, we have revised Section 2.1 of the manuscript to provide detailed information on the chemicals, equipment models, and process parameters used in the fabrication of the ZnO-on-Si resonators. This includes the deposition tools, photolithography equipment, etching systems, photoresist processing conditions, lift-off procedures, and the dimensions of both the rectangular-plate and disk-shaped resonators.

We hope these additions enhance the clarity, transparency, and reproducibility of our fabrication methodology, while fully addressing the reviewer’s request for detailed descriptions of materials, instrumentation, and device geometry.

Please consider changing the subtitle of 2.2. Device operation. Please provide more details about the model used in the simulation. Please explain how the Fig 5. can be used to indicate 1060C does not affect the adjacent devices.

We thank the reviewer for this constructive suggestion. In response, we have revised the subtitle of Section 2.2 from “Device Operation” to “Device Operation and Simulation Model” to better reflect the contents of the section in the revised manuscript.

Additionally, we have expanded the description of the simulation model in Section 2.1 to include details on the material stack employed (constituent materials in the stacked layers), boundary conditions, meshing strategy, applied loads, and analysis type. Specifically, we clarified that the simulation incorporated the full material stack, with resistive Joule heating applied at the anchors and thermal boundary conditions defined to replicate the experimental settings. The mesh was refined locally at the anchors and interfaces to ensure accurate thermal profile and distribution results.

Regarding Figure 5, we have clarified in the revised manuscript that the simulation demonstrates the localized nature of Joule heating — showing that while the annealed resonator reaches a maximum temperature of 1060°C at its anchors, the adjacent resonators on the same substrate remain at significantly lower temperatures (below 70°C), therefore confirming the effectiveness of the localized heating approach in preventing thermal cross-talk or unintentional annealing of neighboring devices. We believe these revisions improve the clarity and completeness of the simulation related explanation and better support the interpretation of results shown in Figure 5.

Please provide some details about how to do the Joule heating in this work.

We thank the reviewer for this constructive comment. In response, we have added details in Section 2.1 explaining that the localized Joule heating was performed by applying a controlled DC current through the resonator anchors using a source meter. The current was gradually increased until the desired temperature (around 1060°C) was reached at the anchors, which is also confirmed by simulations. The applied current ranged between 5 mA and 15 mA with heating times between 30 seconds and 2 minutes, while monitoring voltage and current to avoid overheating. These details have been added in revised manuscript to clarify the experimental procedure and improve the reproducibility of the process.

Please provide some details of the testing of the MEMS.

We thank the reviewer for this important suggestion. In response to this suggestion, we have updated the Experimental Results section to include details on the MEMS testing procedure. The resonators were tested using a Keysight Vector Network Analyzer (VNA) connected via a probe station equipped with Cascade Microtech RF probes. The devices were characterized in air at room temperature, and the motional resistance, resonance frequency, and Q-factor were extracted from the measured admittance curves using equivalent circuit fitting. We believe these added details clarify the experimental testing setup and conditions.

Please provide some details about the TEM testing, such as how to prepare the samples, "100 nm-thick specimens before and after localized annealing"

We thank the reviewer for this great comment. In response, we have clarified in the manuscript that the cross-sectional sample for TEM analysis was prepared by milling the device using a Focused Ion Beam (FIB) system, followed by thinning the sample down to approximately 100 nm to enable high-resolution imaging and interface analysis

Reviewer 3 Report

Comments and Suggestions for Authors

can you add some of your outputs/results in the abstract to make it more attractive for the readers

- figure 14, it is better not to use the table as an image; it is better to b in a table format

- How can you explain the Joule heating effect in this study? It is known that this effect is introduced by the localized plasmonic nanoparticles

- detailes information about the fabrication parameters of the device is needed.

- Figure 3 misses the scale bar

- Figure 5, what is the hotbar indicating? Intensity?

-you really need to compare your results during the discussion section with the current literature reviews

- no scale bar in figure 13

- Table 1, you used a dot and a comma. Please make all in dot

-- also the conclusion is so

Author Response

Reviewer 3:

Can you add some of your outputs/results in the abstract to make it more attractive for the readers

We thank the reviewer for this valuable suggestion. In response, we have revised the abstract to highlight key experimental results, explicitly stating that the Q-factor was enhanced by over 300% (from 916 to 3632) and the insertion loss was reduced by more than 14 dB following the localized Joule heating process. We also emphasized that these improvements were consistently validated by both experimental measurements and simulation results. We believe this update makes the abstract more impactful and insightful, while providing readers with a clear summary of the main outcomes of our work.

- figure 14, it is better not to use the table as an image; it is better to b in a table format

We thank the reviewer for this helpful suggestion. In response, we have reformatted the content originally shown in Figure 14 and created Table 1 in the manuscript as a proper editable table. This format improves the clarity, readability, and consistency with the journal’s guidelines.

- How can you explain the Joule heating effect in this study? It is known that this effect is introduced by the localized plasmonic nanoparticles

We thank the reviewer for highlighting this clarification. We confirm that the Joule heating effect in this study is induced by direct resistive (ohmic) heating of the resonator structure through a controlled DC current applied across its conductive anchors and electrodes. This heating mechanism does not involve plasmonic nanoparticles but instead relies on the intrinsic resistance of the resonator materials.

We believe this explanation is already addressed in Section 2.2 (Device Operation) of the revised manuscript, and we have further clarified this point to prevent any misunderstanding. We would like to thank the reviewer for suggesting us to properly explain the Joule heating effect employed in this work for localized annealing of piezoelectric MEMS resonators.

- Details information about the fabrication parameters of the device is needed.

We thank the reviewer for this helpful suggestion. In response, we have added a detailed summary of the key fabrication parameters used in the device manufacturing process at the end of Section 2.1 of the revised manuscript. This summary includes deposition conditions for the electrodes and ZnO layer, etching parameters, and post-fabrication cleaning steps. We believe this addition enhances the clarity and reproducibility of our fabrication methodology.

- Figure 3 misses the scale bar

We thank the reviewer for bringing this to our attention. In response, we have revised Figure 3 to include the appropriate scale bar, ensuring that the displacement visualization is clearly referenced. We believe this correction improves the clarity and interpretability of the figure.

- Figure 5, what is the hotbar indicating? Intensity?

We thank the reviewer for this clarification request. In response, we have updated the caption of Figure 5 to explicitly state that the color hotbar represents the simulated temperature distribution in degrees Celsius (°C). The updated caption also emphasizes the localized nature of the heating effect achieved through Joule heating. We believe this revision based on the suggestion improves the clarity and interpretation of the figure.

-you really need to compare your results during the discussion section with the current literature reviews

We thank the reviewer for highlighting this key point. In response, we have revised the Results and Discussion section to include a comparison of our experimental findings with recent works reported in the literature on piezoelectric MEMS resonators. Specifically, we have discussed how our achieved Q-factor improvement of over 300% (from 916 to 3632) and the insertion loss reduction by more than 14 dB compare favorably with other reported methods, such as phononic crystal anchoring, mechanical mode coupling, and electrode reconfiguration [4-10].

We emphasized that while previous approaches often rely on complex fabrication steps or structural modifications, our method provides a post-fabrication interface treatment that effectively enhances performance without altering the device geometry. This distinction highlights the practicality and scalability of our approach. We believe this added discussion better positions our work within the current research landscape and addresses the reviewer’s concern.

- no scale bar in figure 13

We thank the reviewer for this valuable observation. In response, we have revised Figure 13 to include the appropriate scale bar, ensuring accurate reference for the imaging results. We believe this correction improves the clarity and scientific rigor of the figure presentation.

- Table 1, you used a dot and a comma. Please make all in dot also the conclusion is so

Thank you for your comments. We have revised it in the manuscript accordingly.

Round 2

Reviewer 1 Report

Comments and Suggestions for Authors

The authors addressed all the reviewer's comments.

Reviewer 3 Report

Comments and Suggestions for Authors

Thank you for your report.